# Trusted Multi-View Classification with Expert Knowledge Constraints

**Xinyan Liang** [1] **Shijie Wang** [1] **Yuhua Qian** [1] **Qian Guo** [2] **Liang Du** [1] **Bingbing Jiang** [3] **Tingjin Luo** [4] **Feijiang Li** [1]

## Abstract

Trusted multi-view classification (TMVC) based on the Dempster-Shafer theory has gained significant recognition for its reliability in safety-critical applications. However, existing methods predominantly focus on providing confidence levels for decision outcomes without explaining the reasoning behind these decisions. Moreover, the reliance on first-order statistical magnitudes of belief masses often inadequately capture the intrinsic uncertainty within the evidence. To address these limitations, we propose a novel framework termed Trusted Multi-view Classification Constrained with Expert Knowledge (TMCEK). TMCEK integrates expert knowledge to enhance feature-level interpretability and introduces a distribution-aware subjective opinion mechanism to derive more reliable and realistic confidence estimates. The theoretical superiority of the proposed uncertainty measure over conventional approaches is rigorously established. Extensive experiments conducted on three multi-view datasets for sleep stage classification demonstrate that TMCEK achieves state-of-the-art performance while offering interpretability at both the feature and decision levels. These results position TMCEK as a robust and interpretable solution for MVC in safety-critical domains. The code is available at https://github.com/jie019/TMCEK_ICML2025.

[1]Institute of Big Data Science and Industry, Key Laboratory of Evolutionary Science Intelligence of Shanxi Province, Shanxi University, Taiyuan, China [2]Shanxi Key Laboratory of Big Data Analysis and Parallel Computing, School of Computer Science and Technology, Taiyuan University of Science and Technology, Taiyuan, China [3]School of Information Science and Technology, Hangzhou Normal University, Hangzhou, China [4]College of Science, National University of Defense Technology, Changsha, China. Correspondence to: Yuhua Qian <jinchengqyh@126.com>.

*Proceedings of the 42nd International Conference on Machine Learning*, Vancouver, Canada. PMLR 267, 2025. Copyright 2025 by the author(s).

## 1. Introduction

Sleep disorder is related to lots of diseases like insomnia, narcolepsy, obstructive sleep apnea syndrome (OSA) (Jahrami et al., 2022). Sleep stage classification (SSC) is the primary diagnostic tool for sleep disorder (Guillot et al., 2020; Wulff et al., 2010). Multi-view learning (MVL) is a powerful paradigm that leverages diverse data representations to improve model performance and robustness (Zhang et al., 2020; Liang et al., 2022; Wei et al., 2025; Yuan et al., 2025), making it particularly valuable in SSC that is mainly based on multi-view polysomnography signals such as EEG, EOG and EMG (Phan et al., 2020). By leveraging data from multiple perspectives, MVL not only enhances diagnostic accuracy but also offers better generalization across diverse patient populations.

While MVL addresses many challenges in SSC, trustworthiness and interpretability remain critical concerns (Wang et al., 2023; Zou et al., 2023). Trusted multi-view learning aims to improve the reliability of predictions by incorporating uncertainty estimation (Han et al., 2023). However, two significant limitations persist: (1) *feature-level opacity*: Current trusted learning methods often function as black boxes at the feature level, failing to clarify which features are critical and how they contribute to the decision-making process. The lack of transparency reduces trust and interpretability for clinicians and patients. (2) *Inaccurate confidence estimates at decision level*: Existing methods primarily rely on the quantity of evidences for uncertainty estimation, without considering their distribution. As a result, confidence estimates may deviate significantly from expected values, particularly in scenarios involving ambiguous or conflicting data, undermining their practical utility.

In this paper, we propose a novel trusted multi-view learning framework that combines interpretability at both the feature and decision levels. Specifically, we firstly utilize Gabor functions in the initial layers of the model to embed expert knowledge into feature extraction, enabling explicit representation of critical features. This approach allows for better understanding of which features contribute to classification decisions, improving transparency and reliance. Second, we improve uncertainty estimation by introducing the distribution of evidence as an additional factor, moving

beyond traditional reliance on the quantity of evidence. By capturing both the magnitude and distribution of evidence, our method provides more reliable confidence estimates, especially in scenarios with ambiguous or conflicting data. These innovations enhance the interpretability, reliability, and robustness of multi-view learning in high-stakes applications like automated sleep stage classification.

Our main contributions are summarized as follows: (1) We propose a novel trusted multi-view learning framework that enhances feature-level interpretability by embedding expert knowledge, enabling explicit identification of critical features contributing to classification decisions. (2) We introduce an improved uncertainty estimation mechanism by incorporating the distribution of evidence, providing more reliable and realistic confidence estimates. (3) Due to the rich expert knowledge available in the sleep domain, we instantiate our method on the sleep datasets, and the experiments show that our approach improves both interpretability and reliability, while also outperforming baseline methods in terms of accuracy.

## 2. Related Work

### 2.1. Trusted Multi-View Learning

Multi-view learning has become a powerful approach for integrating complementary information from multiple data representations, leading to more robust and accurate models across a range of applications (Liang et al., 2024; Wen et al., 2024; Zhang et al., 2024). However, these methods often fail to capture the uncertainty of their predictions. Evidential deep learning (EDL) (Sensoy et al., 2018) uses subjective logic-based approaches to avoid sampling by explicitly modeling uncertainty, making them computationally efficient. Recent work extends EDL into multi-view learning (Liang et al., 2025). A notable approach, trusted multi-view classification (TMC) (Han et al., 2023), employs Dempster's combination rule (Jøsang & Hankin, 2012), assigning lower weights to views with high uncertainty, thereby prioritizing more reliable views. Building on this foundation, several aggregation methods have advanced uncertainty handling by refining how views are integrated (Liu et al., 2023; Zhang et al., 2023). These methods commonly exhibit the property that adding another opinion reduces overall uncertainty. Since data of different views are might not aligned, RCML (Xu et al., 2024) proposes a new aggregation method to ensure that integrating conflicting views appropriately raises uncertainty. Despite these advances, current methods exhibit notable limitations: *feature-level opacity* and *Inaccurate confidence estimates at decision level*. In light of these shortcomings, this work proposes a novel framework to address these gaps by enhancing interpretability at the feature level and introducing a more comprehensive uncertainty estimation method that moves beyond evidence magnitude alone.

### 2.2. Automatic Sleep Stage Classification

Existing work on automatic sleep staging can be categorized three main categories based on the types of signal input representation of the network. The first uses raw one-dimensional (1D) signals directly as input to capture sequential features by one-dimensional convolutional neural networks (1D CNNs) (Chambon et al., 2017), recurrent neural networks (RNNs) (Dong et al., 2016) and attention mechanism (Phan et al., 2018b). The second converts raw signals into two-dimensional (2D) spectrograms using techniques such as the continuous wavelet transform (Kuo et al., 2022) or short-time Fourier transform (STFT) (Guillot et al., 2019). Two-dimensional convolutional neural networks (2D CNNs) are then used to process these spectrograms, which capture essential frequency characteristics linked to each sleep stage. The third combines both temporal and time-frequency representations which employs a dual-stream architecture. In this structure, each branch of the model processes a different view. Then integrating the outputs from each view to produce a fusion output by concatenate operation. XSleepNet (Phan et al., 2020) uses the outputs of three branches to compute losses. SleepPrintNet (Jia et al., 2020) only uses the fusion output to compute losses. Compared with the above methods that focuses on classification performance, our method focuses on trustworthiness and interpretability besides classification performance.

## 3. The Proposed Method

In this study, we propose a method that integrates both time-domain and time-frequency (T-F) domain representations of biosignals to establish a robust multi-view learning framework. We design two subnetworks to independently process the two types of input signals and adopt late fusion at the decision level, where the classification outputs of the two subnetworks are combined to produce the final prediction. The overall architecture of the model is illustrated in Fig. 1.

### 3.1. The Interpretation of Feature Level

Filter parametrization has become a frequently-used technique for designing interpretability deep neural networks (Xie et al., 2023; Zheng et al., 2024). The Gabor function is often used as filters in signal processing tasks (Chang & Morgan, 2014). To explicitly examine the features that contribute to decision-making, we follow the strategy (Niknazar & Mednick, 2024) to parameterize Gabor function, and use them to fit specific signal patterns that align with expert knowledge such as slow oscillations ($\sim$1 Hz), alpha (8–13 Hz), theta (3–7 Hz), spindles (15–18 Hz bursts of activity in a spindle shape), K-complexes (large biphasic waves). Specifically, some Gabor functions are first embedded into

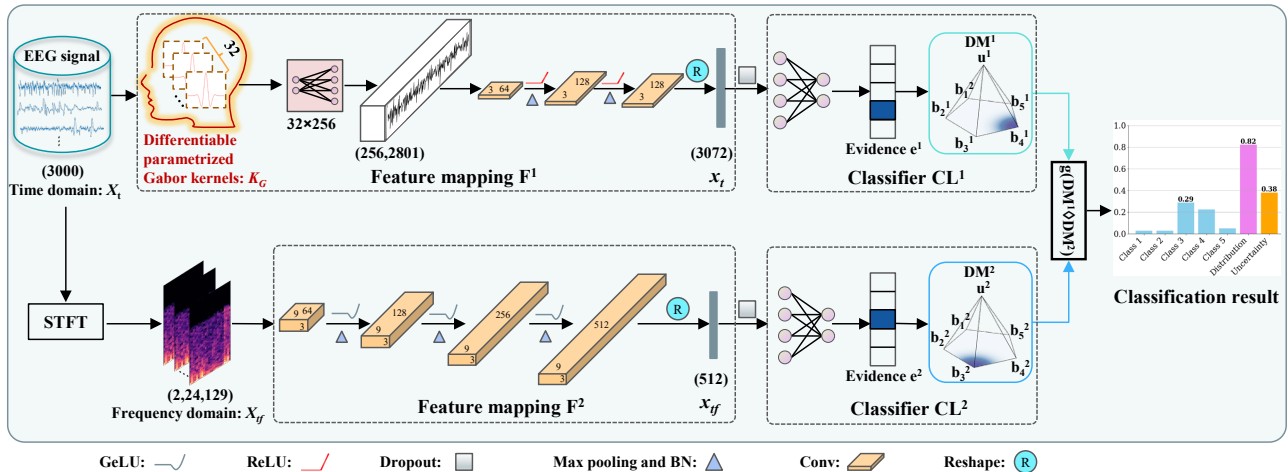

*Figure 1.* The whole framework of model.

the first convolutional layer as its kernels $K_G$, and then a convolution operation is performed between the input signal $X_t$ and $K_G$. The outputs of the convolution operation reflect the correlation between both of them. At the end of the training, the kernel $K_G$ is optimized to be pattern which facilitates decision-making. The process can be mathematically expressed as follows:

$$K_G(T) = e^{-\left(\frac{T-u}{\sigma}\right)^2} \cos\left(2\pi f T\right), \qquad (1)$$

$$Output = K_G * X_t, \qquad (2)$$

where $u$ is center of the kernel function (temporal or spatial location), $\sigma$ is controls the width of the Gaussian envelope, $f$ is frequency of the oscillations in the cosine part of the kernel, $T$ is the time variable used in evaluating the kernel ($-1s \leq T \leq 1s$), and $*$ is convolution operator.

Feature mapping $F^1$ and Classifier $CL^1$. In order to transform a raw signal $X_t$ into a high-level feature $x_t$, the feature mapping $F^1$: $X_t \to x_t$ is realized by convolutional neural networks (CNNs). The output of the Gabor convolutional layer (GCL) represents the magnitude of the specific waveforms (Gabor kernels $K_G$) across the time. Then the resulting Gabor features are used as input to the one-dimensional CNNs to further extract discriminative features. Then flatten and randomly discard some of the processed features to form epoch-wise (30s) feature vector $x_t = F^1(K_G * X_t) \in \mathbb{R}^{3072}$. In the end, through three fully connected layers, we get the probability distribution evidence $e^1 = CL^1(x_t) \in \mathbb{R}^5$ in the time domain.

Feature mapping $F^2$ and Classifier $CL^2$. Simultaneously, the EEG raw signals $X_t$ in the time domain is transformed into the time-frequency (T-F) domain using the Short-Time Fourier Transform (STFT), yielding time-frequency representation $X_{tf} \in \mathbb{R}^{(2 \times T \times F)}$ where $F$ is frequency bins and $T$ is time steps. Same as $F^1$, the feature mapping $F^2$: $X_{tf}$

$\to x_{tf}$ is also realized by convolutional neural networks (CNNs). The processed frequency domain features are reshaped into one-dimensional feature vector by adaptive average pooling. Then the vector $x_{tf} = F^2(X_{tf}) \in \mathbb{R}^{512}$ is passed to three fully connected layers, we get the probability distribution evidence $e^2 = CL^2(x_{tf}) \in \mathbb{R}^5$ in the time-frequency domain. Details on the network architecture are provided in the Appendix A.3.

### 3.2. The Interpretation of Decision Level

In the previous process, we learn view-specific evidence by $F^1$, $CL^1$, $F^2$ and $CL^2$, which could be termed as the amount of support the classification collected from data. Then the view-specific distributions of the class probabilities are modeled by Dirichlet distribution, parameterized with view-specific evidence. From the distributions, we can construct mass consisting of the belief quality of each category and the overall uncertainty. We also combine a conflicting mass aggregation strategy based on trusted fusion to reduce decision conflicts caused by view-specific $F^1$ and $F^2$.

**View-Specific Evidencial Deep Learning.** In decision layer, it is essential to ensure accurate and trustworthy predictions. Traditional methods like softmax layers often overestimate confidence, particularly in incorrect predictions (Wang et al., 2021). Due to their reliance on single-point probability estimates, this limits their ability to capture true model confidence and risk. To overcome the limitation, evidential deep learning (EDL) which is based on evidence theory under the framework of subjective logic (SL) has been introduced (Sensoy et al., 2018). Evidence $e$ here refers to the information extracted from the input data that supports the classification decision, and it is used to derive a belief mass $b_i$ to each class label and an overall uncertainty mass $u$ to the whole frame based on the evidence theory.

**Definition 3.1.** (**Subjective opinion** (Han et al., 2023)): Let $e = [e_1, e_2, \cdots, e_K]$ be the evidences where $e_k$ denotes the $k$-th category evidence. The parameter $\alpha$ of the Dirichlet distribution is defined by $\alpha = e + 1$. Then the subjective opinion can be denoted as $M = [b_1, b_2, \cdots, b_K, u]$ derived by the Dirichlet distribution $\text{Dir}(p|\alpha)$, where $p$ is the class probability vector on a simplex.

For the $v$th view, then the belief mass $b_j^v$ and the uncertainty $u^v$ are computed as: $b_j^v = \frac{e_j^v}{S^v}, u^v = \frac{K}{S^v}$, where $S^v = \sum_{j=1}^{K}(e_j^v + 1) = \sum_{j=1}^{K} \alpha_j^v$ is the Dirichlet strength.

The core of this framework is that the more evidence there is, the higher the quality of belief in a category, and the more confident the model is in predicting that category. Conversely, when there is less evidence, the overall uncertainty increases. The mean of the corresponding Dirichlet distribution $\hat{p}^v$ for the class probability $\hat{p}_j^v$ is computed as $\hat{p}_j^v = \frac{\alpha_j^v}{S^v}$. In subjective opinion, the uncertainty mass is defined as $u^v = \frac{K}{S^v} = \frac{K}{\sum_{j=1}^{K}(e_j^v + 1)}$, which implies that $u$ depends solely on the aggregate sum of evidences $e$. So, it is unsensitive to distribution of evidences $e$. The problem, which is called as *Evidence Distribution-unaware Problem*, can be illustrated using the below example.

**Example 3.2.** Given an input $x$, we feed it into a network to obtain evidence $e_{normal}$=[4,1,1,1,0]. From this, we compute $\alpha_{normal}$=[5,2,2,2,1] and uncertainty $u_{normal}$ =5/12. Now, if we add noise to $x$, the evidence $e_{noisy}$=[2,2,2,1,0], leading to $\alpha_{noisy}$=[3,3,3,2,1] and $u_{noisy}$= 5/12. Normally, as the distribution becomes more concentrated, the uncertainty should increase. Interestingly, after adding noise, the uncertainty $u$ remains constant, which is counterintuitive and clearly unreasonable. We present this problem in Fig. 2, the complete presentation is in Fig. 10 Appendix A.8.

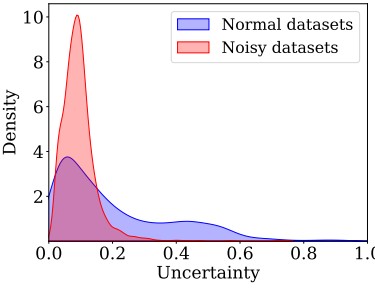

*Figure 2.* Density of uncertainty before and after adding noise.

To overcome the limitation inherent in Definition 3.1, we propose a novel *distribution-aware subjective opinion* framework that extends the conventional model through the incorporation of an evidence distribution concentration measure. Its definition is as follows.

**Definition 3.3.** (**Distribution-aware subjective opinion**): Let $e = [e_1, e_2, \cdots, e_K]$ be the evidences, $d$ denote the

concentration of $e$. A larger $d$ indicates a more concentrated evidence distribution, which corresponds to greater uncertainty. And then distribution-aware subjective opinion is defined as $DM = [b_1, b_2, \cdots, b_K, d, u]$, we redefine the calculation of $b_j^v$ and $u^v$ as follows:

$$b_k = \frac{e_k}{S}, u = \frac{Kd}{S}, d = \frac{(1 + Gini(e))}{2} \qquad (3)$$

where $S = \sum_{j=1}^{K}(e_j + d), Gini(e) = 1 - \sum_{k=1}^{K} p_k^2$ is Gini coefficient where $p_k = \frac{e_k}{\sum_{j=1}^{K} e_j}$ is probability of class $k$.

In contrast to conventional subjective opinion frameworks, our approach determines the uncertainty measure $u$ through a dual consideration of both the cumulative evidence sum and its distribution characteristics, represented by $e$. This enhanced formulation is formally characterized by the theoretical analysis in Subsection 3.3, which demonstrates its superior sensitivity in uncertainty quantification compared to existing methods.

**Evidential Multi-View Fusion via Distributed Mass Aggregation.** After obtaining $V$ independent sets of probability masses assignments $\{DM^v\}_1^V$, where $DM^v = \left[\{b_j^v\}_{j=1}^{K}, d^v, u^v\right]$ under each view, we next need to combine them to obtain a joint mass $DM = \left[\{b_j\}_{j=1}^{K}, d, u\right]$. Misalignment of multi-view data in feature mapping can cause conflicts. We would diminish their impact in the fusion stage.

The joint mass $DM = \left[\{b_j\}_{j=1}^{K}, d, u\right]$ is calculated from the two sets of masses $DM^1 = \left[\{b_j^1\}_{j=1}^{K}, d^1, u^1\right]$ and $DM^2 = \left[\{b_j^2\}_{j=1}^{K}, d^2, u^2\right]$ in the following manner:

$$DM^{1\Diamond2} = DM^1 \Diamond DM^2 = (b^{1\Diamond2}, u^{1\Diamond2}, d^{1\Diamond2}), \quad (4)$$

$$b_j^{1\Diamond2} = \frac{b_j^1 u^2 + b_j^2 u^1}{u^1 + u^2}, u^{1\Diamond2} = \frac{2u^1 u^2}{u^1 + u^2}, d^{1\Diamond2} = \frac{2d^1 d^2}{d^1 + d^2}. \quad (5)$$

The averaging belief fusion can be computed simply by $\frac{d^2 e_k^1 + d^1 e_k^2}{d^1 + d^2}$ in Appendix A.2. We can fusion the final joint mass $DM$ from different views with the following rule:

$$DM = DM^1 \Diamond DM^2 \Diamond \cdots \Diamond DM^v. \qquad (6)$$

Based on the above combination rule, we can obtain the estimated multi-view joint evidence $e$ and the corresponding parameters of joint Dirichlet distribution $\alpha$ to produce the final probability of each class and the overall uncertainty.

**Loss Function.** For instance $\{X_i^v\}_1^V$, $e_i^v = \text{CL}^v(\text{F}^v(X_i^v))$ represent the evidence vector predicted by the network for the classification. To ensure that the network outputs non-negative values, we need to replace the softmax layer of the

traditional neural network based classifier with the activation function layer (ReLU). Different from the typical cross-entropy loss used in traditional neural networks below:

$$\mathcal{L}_{ce} = -\sum_{j=1}^{K} y_{ij} \log p_{ij}, \tag{7}$$

where $p_{ij}$ is the predicted probability of the $i$th sample for class $j$. For our model, we can get the parameter $\boldsymbol{\alpha}_i$ of the Dirichlet distribution by $\boldsymbol{\alpha}_i = \boldsymbol{e}_i + d$. Based on Eq. (7), we have the adjusted cross-entropy loss using evidence-based approach:

$$\mathcal{L}_{ace}(\boldsymbol{\alpha}_i) = \int \left[ \sum_{j=1}^{K} -y_{ij} \log p_{ij} \right] \frac{1}{B(\boldsymbol{\alpha}_i)} \prod_{j=1}^{K} p_{ij}^{\alpha_{ij}-1} d\boldsymbol{p}_i$$
$$= \sum_{j=1}^{K} y_{ij} \left( \psi(S_i) - \psi(\alpha_{ij}) \right), \tag{13}$$

where $\psi(\cdot)$ is the digamma function. The above loss function does not guarantee that the evidence generated by the incorrect labels is lower. To address this issue, we can introduce an additional term in the loss function, namely the Kullback-Leibler (KL) divergence:

$$\mathcal{L}_{KL}(\boldsymbol{\alpha}_i) = KL \left[ D(\boldsymbol{p}_i | \tilde{\boldsymbol{\alpha}}_i) \parallel D(\boldsymbol{p}_i | 1) \right] \tag{8}$$
$$= \log \left( \frac{\Gamma \left( \sum_{j=1}^{K} \tilde{\alpha}_{ij} \right)}{\Gamma(K) \prod_{j=1}^{K} \Gamma(\tilde{\alpha}_{ij})} \right)$$
$$+ \sum_{j=1}^{K} (\tilde{\alpha}_{ij} - 1) \left[ \psi(\tilde{\alpha}_{ij}) - \psi \left( \sum_{j=1}^{K} \tilde{\alpha}_{ij} \right) \right],$$

where $D(\boldsymbol{p}_i | 1)$ is the uniform Dirichlet distribution, $\tilde{\alpha}_i = y_i + (1 - y_i) \odot \alpha_i$ is the Dirichlet parameters after removal of the non-misleading evidence from predicted parameters $\alpha_i$ for the $i$-th instance, and $\Gamma(\cdot)$ is the gamma function.

Therefore, given the Dirichlet distribution with parameter $\alpha_i$ for the $i$-th instance, the loss is:

$$\mathcal{L}_{acc}(\boldsymbol{\alpha}_i) = \mathcal{L}_{ace}(\boldsymbol{\alpha}_i) + \lambda_t \mathcal{L}_{KL}(\boldsymbol{\alpha}_i), \tag{9}$$

where $\lambda_t = \min(1.0, t/T) \in [0, 1]$ is the annealing coefficient, $t$ is the index of the current training epoch, and $T$ is the annealing step. By gradually increasing the influence of KL divergence in loss, premature convergence of misclassified instances to uniform distribution can be avoided.

In order to ensure the consistency of results between different mass during training, minimizing the conflicts between mass was adopted. The consistency loss for the instance $\{x_i^v\}_{v=1}^V$ is calculated as (Xu et al., 2024):

$$\mathcal{L}_{con1} = \sum_{m=1}^{V} \sum_{n \neq m}^{V} \left( \frac{\sum_{j=1}^{K} |p_j^m - p_j^n| \cdot (1 - u^m) \cdot (1 - u^n)}{2 \cdot (V-1)} \right),$$

$$\mathcal{L}_{con2} = \frac{1}{V(V-1)} \sum_{m=1}^{V-1} \sum_{n=m+1}^{V} \frac{\boldsymbol{e}^m \cdot \boldsymbol{e}^n}{\|\boldsymbol{e}^m\| \|\boldsymbol{e}^n\|},$$
$$\mathcal{L}_{con} = \zeta \mathcal{L}_{con1} + \eta \mathcal{L}_{con2}. \tag{10}$$

To sum up, the overall loss function for a specific instance $\{X_i^v\}_{v=1}^V$ can be calculated as:

$$\mathcal{L} = \mathcal{L}_{acc}(\boldsymbol{\alpha}_i) + \beta \sum_{v=1}^{V} \mathcal{L}_{acc}(\boldsymbol{\alpha}_i^v) + \gamma \mathcal{L}_{con}. \tag{11}$$

### 3.3. Theoretical Analysis

To demonstrate the superiority of the distribution-aware subjective opinion framework, we conduct a comprehensive theoretical analysis. This examination reveals several key advantages: (1) enhanced modeling capability for uncertainty quantification through explicit distribution consideration, (2) the relation between distribution-aware subjective opinion aggregation and evidence aggregation, and (3) aggregation properties. The theoretical framework establishes a rigorous mathematical foundation that not only justifies its practical effectiveness but also provides insights into its relationship with conventional subjective logic approaches.

**Proposition 3.4.** *Given two evidences* $\boldsymbol{e}^1 = [e_1^1, e_2^1, \cdots, e_K^1]$ *and* $\boldsymbol{e}^2 = [e_1^2, e_2^2, \cdots, e_K^2]$. *If* $\sum_j^K e_j^1 = \sum_j^K e_j^2$ *and* $d(\boldsymbol{e}^1) \leq d(\boldsymbol{e}^2)$, *then* $u^1 \leq u^2$.

Proposition 3.4 establishes that our modified opinion framework properly captures the uncertainty quantification of evidence through its dispersion characteristics. The complete mathematical proof is provided in Appendix A.1.

**Proposition 3.5.** *The distribution-aware subjective opinion aggregation operation* $DM^{1 \diamond 2} = DM^1 \diamond DM^2$ *is mathematically equivalent to the weighted evidence pooling:*

$$e^{1 \diamond 2} = \frac{d^2 e^1 + d^1 e^2}{d^1 + d^2}. \tag{12}$$

Proposition 3.5 demonstrates that the proposed distribution-aware aggregation mechanism can be effectively implemented through a dispersion-weighted evidence pooling scheme. The detailed proof is available in Appendix A.2.

**Proposition 3.6.** *Let* $\boldsymbol{DM}^1 = [b_1^1, b_2^1, \cdots, b_K^1, d^1, u^1]$ *and* $\boldsymbol{DM}^2 = [b_1^2, b_2^2, \cdots, b_K^2, d^2, u^2]$ *represent distribution-aware subjective opinions from two distinct views, with* $u^1 < u^2$. *The aggregation process exhibits the following properties:*

- *When* $\boldsymbol{DM}^1$ *is aggregated into* $\boldsymbol{DM}^2$, *the resulting uncertainty mass decreases:* $u^2_{new} < u^2$;

- *When* $\boldsymbol{DM}^2$ *is aggregated into* $\boldsymbol{DM}^1$, *the resulting uncertainty mass increases:* $u^1_{new} > u^1$.

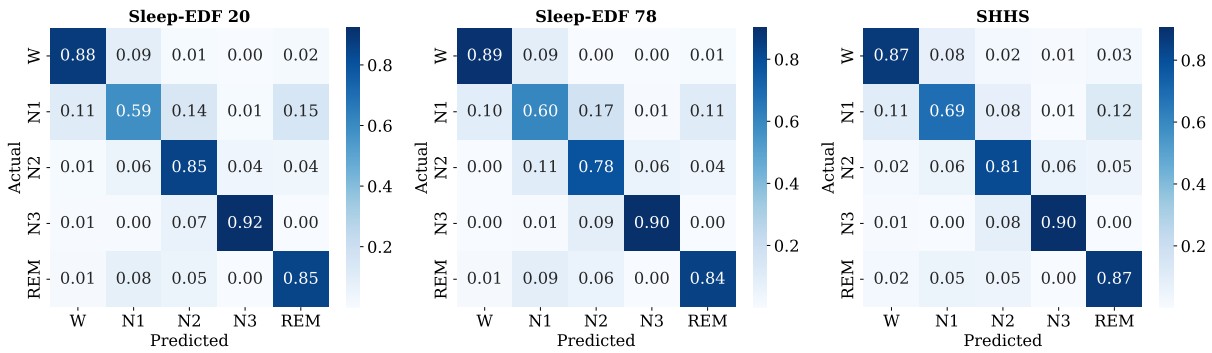

*Figure 3.* The normalized confusion matrix.

Proposition 3.6 reveals that the proposed aggregation method naturally accounts for potential conflicts between different opinions through its uncertainty-aware fusion mechanism. Based on Eq. 5, its proof is obvious.

## 4. Experiments

### 4.1. Experimental Setups

**Datasets.** In this experiments, we use three public datasets including Sleep-EDF 20, Sleep-EDF 78 and Sleep Heart Health Study (SHHS) as shown in Appendix A.4. For each dataset, we use a single EEG channel for various models in our experiments.

**Compared Methods.** We compared our model with the several representative methods on three datasets including DeepSleepNet (Supratak et al., 2017), ARNN+SVM (Phan et al., 2018b), SleepEEGNet (Mousavi et al., 2019), ResNetLSTM (Sun et al., 2018), MultiTaskCNN (Phan et al., 2018a), DFSC (Liu et al., 2018), ResAtten (Qu et al., 2020), AttnSleep (Eldele et al., 2021) and MISC (Niknazar & Mednick, 2024). For a detailed description of these methods, please refer to Appendix A.5.

**Evaluation Metrics and Implementation Details.** To evaluate the performance of the proposed method of sleep stage scoring. We used accuracy (Acc), macro F1-score (MF1), and Cohen's kappa (Kappa). Among these measures, F1-score showed the performance of the method with respect to each sleep stage separately. The experimental setups are detailed in Appendix A.6

### 4.2. Experimental Results

In the section, we conduct the following experiments to evaluate our model from three aspects: performance comparison, confusion matrix analysis and hypnogram visualization.

**Performance Comparison.** To verify the superiority of the proposed sleep scoring system, we compare the three evaluation metrics of our model with the baseline on two

*Table 1.* Results comparison between the proposed method and some other deep learning methods.

| DATASET | STUDY | ACC | MF1 | KAPPA |
|---|---|---|---|---|
| | DEEPSLEEPNET | 81.9 | 76.6 | 0.760 |
| | ARNN+SVM | 79.1 | 69.8 | 0.700 |
| | MULTITASKCNN | 83.1 | 75.0 | 0.77 |
| EDF20 | DFSC | 84.44 | 78.25 | 0.784 |
| | RESATTEN | 84.3 | 79.0 | 0.78 |
| | MISC | 81.9 | 74.4 | 0.75 |
| | **TMCEK (OURS)** | **85.0** | **80.2** | **0.80** |
| | DEEPSLEEPNET | 77.8 | 73.9 | 0.73 |
| | SLEEPEEGNET | 74.2 | 69.6 | 0.66 |
| | RESNETLSTM | 78.9 | 71.4 | 0.71 |
| EDF78 | MULTITASKCNN | 79.6 | 72.8 | 0.72 |
| | ATTNSLEEP | 81.3 | 75.1 | 0.74 |
| | MISC | 77.4 | 69.8 | 0.68 |
| | **TMCEK (OURS)** | **81.4** | **77.5** | **0.75** |
| | DEEPSLEEPNET | 81.0 | 73.9 | 0.73 |
| | SLEEPEEGNET | 73.9 | 68.4 | 0.65 |
| | RESNETLSTM | 83.3 | 69.4 | 0.76 |
| SHHS | MULTITASKCNN | 81.4 | 71.2 | 0.74 |
| | ATTNSLEEP | 84.2 | 75.3 | 0.78 |
| | MISC | 79.1 | 72.6 | 0.71 |
| | **TMCEK (OURS)** | **84.3** | **78.0** | **0.79** |

datasets. The evaluation metrics include Acc, MF1 and Kappa. The rxperimental results on the three datasets are reported in Table 1, one can draw the following conclusions: (1) TMCEK achieves the best results in terms of all metrics. This indicates that the proposed method which deeply integrates time-domain and frequency-domain features effectively improve performance. (2) Compared with methods which focus on either temporal or frequency modeling, our method achieved better results, which further verified this effectivity of fusing complementary information. (3) Compared to DFSC, our superior performance may be attributed to fusion method leverages preliminary feature interaction and post-fusion, ensuring deeper integration of complementary information.

**Confusion Matrix Analysis.** The classification results of the confusion matrix are shown in Fig. 3. From the results,

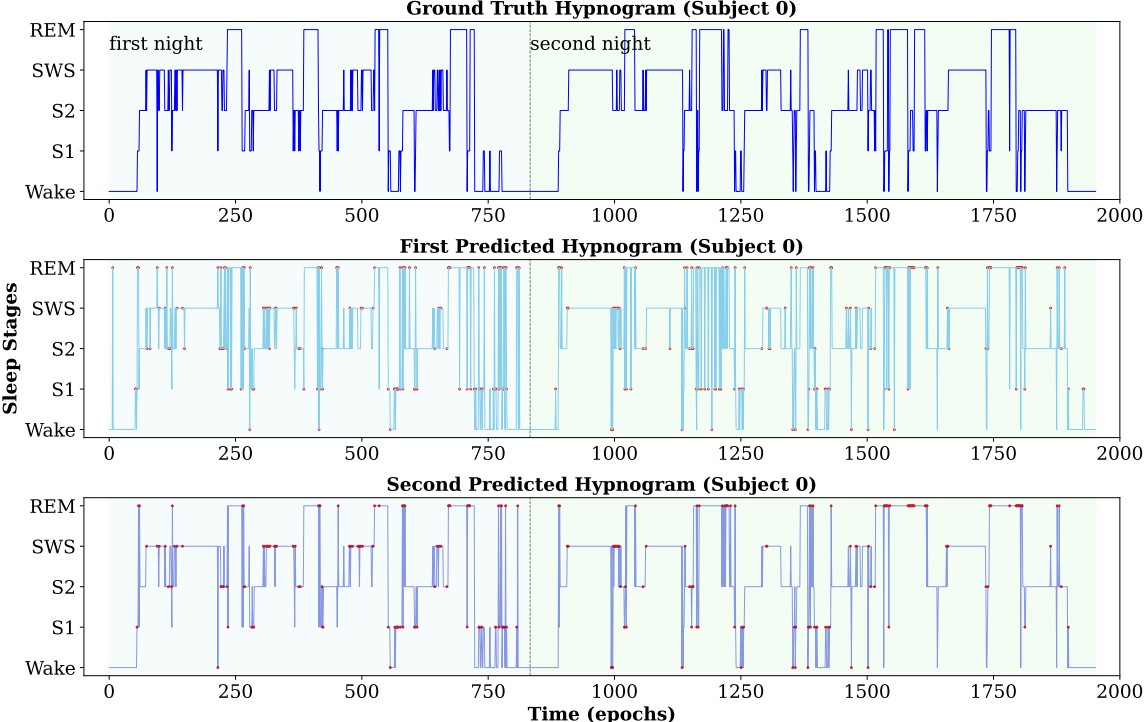

*Figure 4.* Example of ground truth (expert's scoring) and predicted hypnogram for one fold. The red dots indicate that the network misclassified the 30 s epoch.

it can be observed that TMCEK has the best classification results on W and N3 stages. The classification performance of the N1 stage is the worst among all sleep stages. The reasons of the results may be that the N1 stage, as a transitional state between wakefulness and deeper sleep stages like N2, exhibits overlapping features with both. For example, EEG characteristics of N1, such as low-amplitude theta waves and occasional alpha waves, are also present in wakefulness and N2, leading to blurred boundaries. This feature ambiguity makes it difficult for traditional feature extraction methods and deep learning models to distinguish N1 from other stages, particularly W and N2. In addition, the N1 stage is typically underrepresented in sleep staging datasets: N1 accounts for approximately 5% of a night's sleep, much lower than other stages like N2 or REM. This imbalance biases classification models towards dominant stages, thereby degrading the performance for the N1 stage.

**Hypnogram Visualization.** In Fig. 4 we present an original manually scored hypnogram and its corresponding estimated sleep hypnogram using the trained single-epoch and multi-epoch networks for one fold on the Sleep-EDF 20. Its score is approximately equal to the mean score across the entire dataset. From Fig. 4, it can be observed that there are many misclassified points in the single epoch network output. Using the multi-epoch network to model transition rules between epochs can eliminate partial misclassified points

and increased the classification performance significantly.

### 4.3. Interpretation

To increase the system's interpretability and reliability, we introduce the Gabor kernel at the first convolution layer in time domain. In addition, we can know whether the decision is credible by uncertainty estimation. Next, we analyze these two aspects respectively.

#### 4.3.1. INTERPRETABILITY AT FEATURE-LEVEL

In terms of interpretability, the trainable Gabor kernels in the first layer are used to learn meaningful waveform patterns that are directly associated with sleep stages. These kernels are adjusted during training to capture representative time-frequency structures within the EEG signals. The outputs of this Gabor layer reflect how prominently each learned waveform appears in the input, effectively highlighting characteristic features relevant to sleep staging. The calculation of the overall qualitative impact of each kernel waveform is given in Appendix A.7.

In Fig. 5, we show the top eight EEG optimized Gabor waveforms along with their frequency domain by overall importance (the complete figure is shown in Fig. 8 in Appendix A.8). Some of the optimized kernels were perfectly matched to well-known EEG waveforms including slow waves (SW,

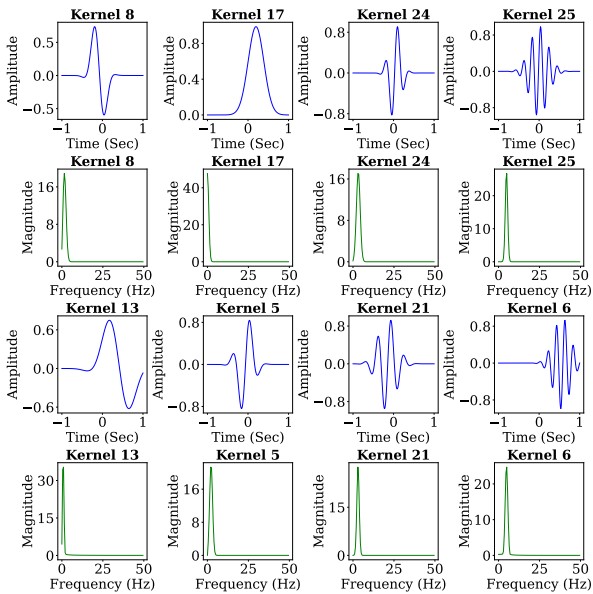

*Figure 5.* Waveform and frequency domains of important optimized Gabor kernels.

1Hz), deta waves(1 to 4Hz), theta waves(3 to 8Hz). Optimized Gabor kernels 8 and 17 is similar to SW and deta waves, kernels 5, 24 and 25 are fitted to theta waves.

Fig. 9 in Appendix A.8 displays the importance of the corresponding Gabor waveforms in the overall sleep staging process and different sleep stages of the single-epoch network. The results in this figure are compatible to the experts' knowledge and the sleep scoring manuals. For example, Gabor kernels 8 and 17 which represent SW and deta waves have highest impact in stage S2 and SWS. On the other hand, the results in Fig. 9 show that some Gabor kernels are not important because the training process could not optimize them or they had redundant information, and other optimized kernels produced enough information for decision making. To improve kernel optimization, our subsequent work considers to apply diversity regularization for explicitly penalizing similarity among kernels and encouraging each to capture distinct patterns.

### 4.3.2. UNCERTAINTY ESTIMATION AT DECISION LEVEL

To further evaluate the estimated uncertainty, we visualize the distribution of normal and noisy data on the Sleep-EDF 20 dataset in Fig. 11 in Appendix A.8. To construct noisy test sets, we introduce Gaussian noise with standard deviation $\sigma = 10, 30, 50, 100$ to of the test instances. The experimental results are presented in Fig. 6. The results reveal that, when the noise intensity is low ($\sigma = 10$), the distribution curve of the noisy instances closely aligns with that of the normal instances. However, as the noise intensity increases, the uncertainty of the noisy instances also increases. This finding indicates that the estimated uncertainty is cor-

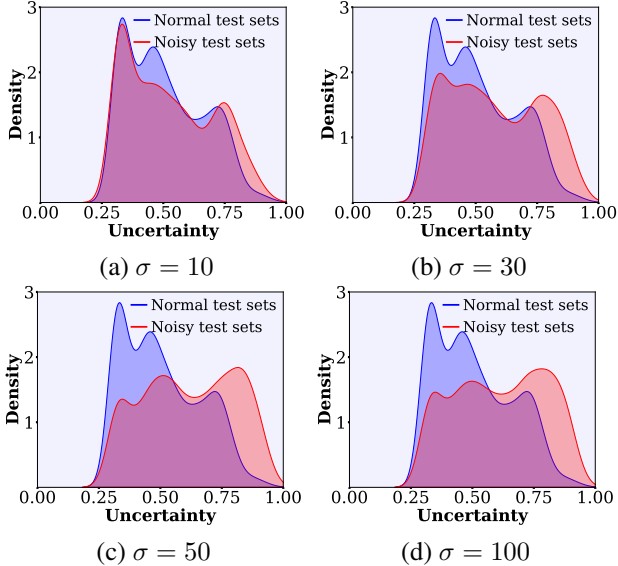

*Figure 6.* Density of uncertainty.

related with the quality of the instances, thereby validating the capability of our method in uncertainty estimation.

### 4.4. Robustness

The robustness of the model can be improved through trusted learning. In order to verify the robustness improvement brought by trusted learning, we select the first fold (the first subject is used as the test set, and the others are used as the training set) for robustness testing on Sleep-EDF 20. We compare the evaluation indicators of the model trained with trusted learning and the model without trusted learning by adding the Gaussian noise with different levels of standard deviations ($\delta$) to time domain view on the test set. From Fig. 7, it can be observed that models using trusted learning are more robust than those without it, which highlights the importance of the reliability of decision results.

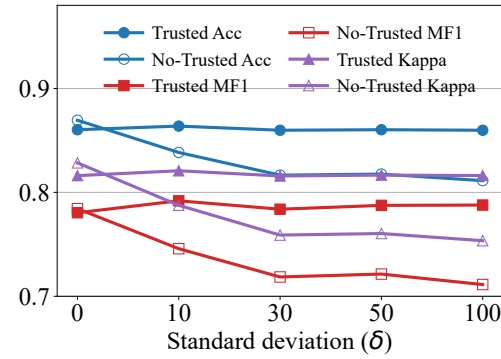

*Figure 7.* Comparison on Sleep-EDF 20 with different noise levels.

### 4.5. Comparison with Trusted Multi-view Methods

To validate the effectiveness of the proposed trustworthy multi-view learning method, we conducted experiments on several benchmark multi-view datasets, including the Hand-

Table 2. Classification accuracy (%) on different datasets. $^*$ indicates results from CCML (Liu et al., 2024).

| DATA | EDL$^*$ | DCCAE$^*$ | CAML$^*$ | ETMC$^*$ | RCML$^*$ | CCML | TMCEK (OURS) |
|---|---|---|---|---|---|---|---|
| HANDWRITTEN | $97.00 \pm 0.16$ | $97.05 \pm 0.24$ | $98.10 \pm 0.12$ | $98.32 \pm 0.22$ | $98.70 \pm 0.19$ | $98.75 \pm 0.27$ | $\mathbf{98.80 \pm 0.76}$ |
| SCENE15 | $60.60 \pm 0.13$ | $64.26 \pm 0.42$ | $70.17 \pm 0.13$ | $66.87 \pm 0.29$ | $71.28 \pm 0.32$ | $72.60 \pm 0.87$ | $\mathbf{74.84 \pm 0.39}$ |
| CUB | $89.51 \pm 0.24$ | $85.39 \pm 1.36$ | $94.33 \pm 0.73$ | $91.05 \pm 0.63$ | $93.28 \pm 2.75$ | $94.58 \pm 1.30$ | $\mathbf{94.67 \pm 1.55}$ |
| PIE | $87.99 \pm 0.56$ | $81.96 \pm 1.04$ | $93.38 \pm 0.80$ | $93.82 \pm 0.82$ | $93.89 \pm 2.46$ | $94.56 \pm 1.83$ | $\mathbf{97.06 \pm 1.80}$ |

Written (HW) and Scene15, CUB and PIE datasets (details in Appendix A.4). We compare our methods with EDL (Sensoy et al., 2018), DCCAE (Wang et al., 2015), CALM (Zhou et al., 2023), ETMC (Han et al., 2023), RCML (Xu et al., 2024) and CCML (Liu et al., 2024). Among them, EDL, ETMC, RCML, and CCML are four widely trusted multi-view methods. Detailed descriptions of the compared methods and implementation specifics are provided in Appendix A.5 and A.6. From Table 2, one can get that TM-CEK achieves the best classification accuracy across all datasets. To further demonstrate the superiority of TMCEK, we select a sample from Scene15 and show its classification performance and uncertainty estimate before and after adding noise. The result is illustrated in Fig. 12. From Fig. 12, it can be observed that (1) RCML misclassifies the noisy sample, whereas our method correctly classifies it. (2) RCML shows a decrease in uncertainty after noise injection due to an overall increase in the amount of evidence. In contrast, TMCEK provides a more accurate uncertainty estimation. This improvement stems from our method's novel dual consideration of both the cumulative evidence and its distributional properties during uncertainty quantification.

## 5. Conclusion

In this paper, we have presented a novel trusted multi-view classification framework constrained with expert knowledge to address critical challenges in trusted multi-view learning. TMCEK effectively integrates expert knowledge by embedding Gabor kernels into the feature extraction module, thereby achieving interpretability at the feature level. This design improves the transparency of the decision-making process, making it more understandable and trustworthy for clinicians and patients. We further improved the reliability of uncertainty estimation by introducing a novel approach that considers not only the quantity but also the distribution of evidence, enhancing trustworthiness in high-stakes medical applications. We theoretically proved that it can enable more precise uncertainty estimation. Furthermore, experimental results on multiple datasets validated the effectiveness of TMCEK, confirming its superior performance in terms of accuracy, interpretability and reliability for multi-view classification tasks. In the future, we focus on the embedded strategies of the expert knowledge from other domain into TMCEK.

## Acknowledgements

This work was supported by National Natural Science Foundation of China (Nos. 62306171, T2495251, 62406218, 62376281, T2495253), the Science and Technology Major Project of Shanxi (No. 202201020101006), and Fundamental Research Program of Shanxi Province (No. 202203021222183).

## Impact Statement

The results demonstrate that our framework bridges the gap between interpretability and performance, making it a promising tool for sleep disorder diagnosis and other healthcare applications where trust and transparency are paramount. Future work could extend this approach to broader domains, further enhancing its utility in diverse multimodal diagnostic settings.

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

# A. Appendix

In the supplemental material:

- **A.1.** Proof of Proposition 3.4.

- **A.2.** Proof of Proposition 3.5.

- **A.3.** Sleep Model Architecture.

- **A.4.** Datasets.

- **A.5.** The Compared Methods.

- **A.6.** The Implementation Details and Evaluation Metrics.

- **A.7.** The Quantification of Gabor Kernel Influence.

- **A.8.** Supplementary Figures.

## A.1. Proof of Proposition 3.4

*Proof.*

$$
\begin{aligned}
u^1 - u^2 &= \frac{Kd^1}{\sum_{j=1}^{K}(e_j^1 + d^1)} - \frac{Kd^2}{\sum_{j=1}^{K}(e_j^2 + d^2)} \\
&= \frac{Kd^1 \sum_{j=1}^{K}(e_j^2 + d^2) - Kd^2 \sum_{j=1}^{K}(e_j^1 + d^1)}{\sum_{j=1}^{K}(e_j^1 + d^1)\sum_{j=1}^{K}(e_j^2 + d^2)} \\
&= \frac{K(d^1 \sum_{j=1}^{K} e_j^2 + d^1 \sum_{j=1}^{K} d^2 - d^2 \sum_{j=1}^{K} e_j^1 - d^2 \sum_{j=1}^{K} d^1)}{\sum_{j=1}^{K}(e_j^1 + d^1)\sum_{j=1}^{K}(e_j^2 + d^2)} \\
&= \frac{K(d^1 \sum_{j=1}^{K} e_j^2 + d^1 Kd^2 - d^2 \sum_{j=1}^{K} e_j^1 - d^2 Kd^1)}{\sum_{j=1}^{K}(e_j^1 + d^1)\sum_{j=1}^{K}(e_j^2 + d^2)} \\
&= \frac{K(d^1 \sum_{j=1}^{K} e_j^2 - d^2 \sum_{j=1}^{K} e_j^1)}{\sum_{j=1}^{K}(e_j^1 + d^1)\sum_{j=1}^{K}(e_j^2 + d^2)} \\
&= \frac{K(d^1 - d^2) \sum_{j=1}^{K} e_j^1}{\sum_{j=1}^{K}(e_j^1 + d^1)\sum_{j=1}^{K}(e_j^2 + d^2)} \quad (\because \sum_{j=1}^{K} e_j^2 = \sum_{j=1}^{K} e_j^1) \\
&= \frac{K(\frac{1+Gini(\boldsymbol{e^1})}{2} - \frac{1+Gini(\boldsymbol{e^2})}{2}) \sum_{j=1}^{K} e_j^1}{\sum_{j=1}^{K}(e_j^1 + d^1)\sum_{j=1}^{K}(e_j^2 + d^2)} \\
&= \frac{K(Gini(\boldsymbol{e^1}) - Gini(\boldsymbol{e^2})) \sum_{j=1}^{K} e_j^1}{2\sum_{j=1}^{K}(e_j^1 + d^1)\sum_{j=1}^{K}(e_j^2 + d^2)} \leq 0 \quad (\because Gini(\boldsymbol{e^1}) \leq Gini(\boldsymbol{e^2}))
\end{aligned}
$$

Hence, $u^1 \leq u^2$ holds. $\qquad\square$

## A.2. Proof of Proposition 3.5

*Proof.* Based on Eqs. 3 and 5, we have

$$b_k = \frac{e_k}{S}, \quad u = \frac{Kd}{S}$$

$$b_k = \frac{b_k^1 u^2 + b_k^2 u^1}{u^1 + u^2}$$

$$u = \frac{2u^1 u^2}{u^1 + u^2}$$

$$d = \frac{2d^1 d^2}{d^1 + d^2}$$

Hence,

$$
\begin{aligned}
e_k &= b_k S \quad (\because b_k = \frac{e_k}{S}) \\
&= \frac{b_k K d}{u} \quad (\because u = \frac{Kd}{S}) \\
&= \frac{b_k^1 u^2 + b_k^2 u^1}{u^1 + u^2} \cdot \frac{u^1 + u^2}{2u^1 u^2} \cdot \frac{2Kd^1 d^2}{d^1 + d^2} \quad (\because b_k = \frac{b_k^1 u^2 + b_k^2 u^1}{u^1 + u^2}, u = \frac{2u^1 u^2}{u^1 + u^2}, d = \frac{2d^1 d^2}{d^1 + d^2}) \\
&= \frac{Kd^1 d^2}{d^1 + d^2} \cdot \frac{b_k^1 u^2 + b_k^2 u^1}{u^1 u^2} \\
&= \frac{Kd^1 d^2}{d^1 + d^2} \cdot \left( \frac{\frac{Ke_k^1 d^2}{S^1 S^2} + \frac{Ke_k^2 d^1}{S^1 S^2}}{\frac{Kd^1}{S^1} \frac{Kd^j}{S^2}} \right) \\
&= \frac{d^2 e_k^1 + d^1 e_k^2}{d^1 + d^2}
\end{aligned}
$$

$\square$

## A.3. Sleep Model Architecture

**Single-Epoch Network.** The proposed model consists of two main modules: the time-domain module and the frequency-domain module. The time-domain module processes raw signals through a series of one-dimensional convolutional layers. These include five convolutional layers with kernel size 3, gradually increasing the number of filters (64, 128, 128, 256, 256) and applying strides to reduce temporal resolution while maintaining key features. Batch normalization is applied after each layer, and ReLU activation is used throughout. The output is flattened, and three fully connected layers with dropout (p=0.6) are employed to map features to a final probability distribution. In the frequency-domain module, the raw signals are first transformed into time-frequency representations $X_{tf}$ using Short-Time Fourier Transform (STFT). The window size is set as 256, with 50% overlap between adjacent frames. 256-point FFT is utilized, leading 256 to 129-D in the feature axis. These representations are passed through convolutional layers with wider kernels (3×9) to capture complex spectral patterns. Filters progressively increase from 64 to 512, with strides along the frequency axis to ensure real-time processing capability. Adaptive average pooling is applied to reshape the features, producing one-dimensional feature vectors for classification. This design leverages complementary information from both time and frequency domains, enhancing feature extraction and ensuring robust and accurate classification.

**Multi-Epoch Network.** As the consideration of transition rules between epochs can effectively improve the performance of sleep scoring (Supratak et al., 2017), we use the multi-epoch network to analyze the inter-epoch temporal context. We obtain the classification results of each epoch ($O_1$ to $O_n$) through the above network. Based on the experts suggestion (Iber et al., 2007), we take ten epochs as input of the multi-epoch network ($O_{n-4}$ to $O_n$ and $O_n$ to $O_{n+4}$). The multi-epoch network consists of two long short-term memory (LSTM) layers and a fully connected layer (FC) for classification. The first LSTM feeds the output of the first level network in the forward direction and the second LSTM feeds the output of the first level network in the backward direction. The LSTM layers in the multi-epoch network consist of two layers with a hidden state size of 10. We use the cross-entropy loss function to train the multi-epoch network.

*Table 3.* Details of three datasets used in our experiments(each sample is a 30-second epoch).

| DATASETS | SUBJECTS | EEG CHANNEL | SAMPLING RATE | W | N1 | N2 | N3 | REM | TOTAL |
|---|---|---|---|---|---|---|---|---|---|
| EDF20 | 20 | FPZ-CZ | 100 Hz | 8285 | 2804 | 17799 | 5703 | 7717 | 42308 |
| EDF78 | 78 | FPZ-CZ | 100 Hz | 65951 | 21522 | 69132 | 13039 | 25835 | 195479 |
| SHHS | 329 | C4-A1 | 125 Hz | 46319 | 10304 | 142125 | 60153 | 65953 | 324854 |

### A.4. Datasets

**Multi-view Datasets. HandWritten**[1] comprises 2000 instances of handwritten numerals ranging from '0' to '9', with 200 patterns per class. It is represented using six feature sets. **Scene15**[2] includes 4485 images from 15 indoor and outdoor scene categories. We extract three types of features HOG, LBP, and GIST. **CUB**[3] consists of 11788 instances associated with text descriptions of 200 different categories of birds, we focus on the first 10 categories and extract image features using GoogleNet and corresponding text features using doc2vec. **PIE**[4] contains 680 facial instances belonging to 68 classes. We extract intensity, LBP, and Gabor as three views.

**Sleep Datasets.** We used three public datasets, namely, Sleep-EDF 20, Sleep-EDF 78 and Sleep Heart Health Study (SHHS) as shown in Table 3.

1) The Sleep-EDF dataset, sourced from PhysioBank (Goldberger et al., 2000), includes two subsets: Sleep-EDF 20 (EDF-20) and Sleep-EDF 78 (EDF-78). Sleep-EDF 20 contains data from 20 subjects, while Sleep-EDF 78 extends this to 78 subjects. These datasets originate from two distinct studies. The first study, Sleep Cassette (SC* files), examines the effects of aging on sleep and involves healthy participants aged 25 to 101 years. The second study, Sleep Telemetry (ST* files), investigates the impact of temazepam on sleep, focusing on 22 Caucasian males and females who were not on any other medications. For both studies, each polysomnography (PSG) file includes two EEG channels (Fpz-Cz, Pz-Oz) sampled at 100 Hz, as well as one EOG channel and one chin EMG channel. We select the Fpz-Cz channel as the input for our experimental models.

2) The SHHS dataset (Zhang et al., 2018), (Quan et al., 1997) is a large-scale, multi-center cohort study designed to explore the cardiovascular and other health impacts of sleep-disordered breathing. Participants in this study presented a range of medical conditions, including pulmonary and cardiovascular diseases. To reduce the influence of these conditions, we followed the subject selection criteria outlined in (Fonseca et al., 2017), focusing on individuals with relatively normal sleep patterns (Apnea Hypopnea Index or AHI below 5). This resulted in a subset of 329 participants from the original pool of 6,441 subjects. For this dataset, we utilized the C4-A1 EEG channel with a sampling rate of 125 Hz.

Across all datasets, the following preprocessing steps were applied. First, any UNKNOWN stages not corresponding to specific sleep stages were excluded. Second, N3 and N4 stages were merged into a single N3 stage following the AASM standard. Finally, only 30 minutes of wakefulness before and after the primary sleep periods were retained to better emphasize the sleep stage classification. We use per-subject 20-fold cross validation, dividing the subjects in each dataset into 20 groups. The recordings in one group were considered as test data, and the rest were used as training data. This process was repeated until all folds were iterated.

### A.5. The Compared Methods

**Multi-view Methods.** There are six compared multi-view methods:

- EDL (Sensoy et al., 2018) quantifies classification uncertainty by placing a Dirichlet distribution on the class probabilities. It models the evidence for each class and uses this evidence to compute the Dirichlet distribution parameters. The Dirichlet distribution provides a measure of uncertainty, which is used to make more reliable predictions.

- DCCAE (Wang et al., 2015) is a deep multi-view representation learning method that uses autoencoders to learn a common representation across multiple views. It maximizes the correlation between the views to extract shared

---

[1] https://archive.ics.uci.edu/dataset/72/multiple+features
[2] https://figshare.com/articles/dataset/15-Scene_Image_Dataset/7007177/1
[3] https://www.vision.caltech.edu/visipedia/CUB-200.html
[4] http://www.cs.cmu.edu/afs/cs/project/PIE/MultiPie/Home.html

information.

- CALM (Zhou et al., 2023) is an enhanced encoding and confidence evaluating framework for trustworthy multi-view classification. It combines early and late fusion strategies to leverage the complementarity of multiple views and improve classification reliability.

- ETMC (Han et al., 2023) extends the Trusted Multi-view Classification (TMC) method by introducing a pseudo-view to enhance interaction between different views. It dynamically evaluates the quality of different views and makes trusted decisions based on uncertainty.

- RCML (Xu et al., 2024) proposes a conflictive opinion aggregation strategy to handle conflictive multi-view data. It ensures the consistency of results between different views during training and quantifies the conflictive degree between views.

- CCML (Liu et al., 2024) constructs view-specific evidential DNNs to learn view-specific evidence. It dynamically decouples the consistent and complementary evidence and processes them according to different principles.

**Sleep Staging Methods.** We compared our model with the following nine baselines:

- DeepSleepNet (Supratak et al., 2017) is a model based on a two-stage neural network architecture, combining 1D CNNs and Bi-LSTMs. It extracts time-frequency features through 1D CNNs and then uses Bi-LSTMs for sequence modeling, and is good at processing long-term sleep stage data.

- ARNN+SVM (Phan et al., 2018b) utilizes bidirectional RNN with an attention mechanism to extract sequential features from EEG, combined with an SVM classifier for sleep staging. Performance is enhanced through learned filter banks.

- MultitaskCNN (Phan et al., 2018a) first converts the original EEG signal into a power spectrum image, and then optimizes the main task (classification) and the related auxiliary task (prediction) by sharing the feature extraction layer, thereby improving the generalization ability of the model.

- DFSC (Liu et al., 2018) applies diffusion geometry to fuse EEG raw signal and spectral information for sleep dynamics visualization and stage prediction, with automatic annotation using SVM.

- ResAtten (Qu et al., 2020) uses CNNs to extract multi-band features and employs the multi-head attention module of the Transformer to model global temporal context, achieving efficient sleep staging.

- SleepEEGNet (Mousavi et al., 2019) focuses on processing specific features of EEG signals. It uses 1D CNNs combined with lightweight design to achieve efficient sleep staging in environments with limited device resources.

- ResnetLSTM (Sun et al., 2018) combines residual networks (ResNet) and LSTM, first extracting spatial features through ResNet and then using LSTM for time series modeling, thereby improving the accuracy and robustness of model.

- AttnSleep (Eldele et al., 2021) first extracts and optimizes features through a multi-resolution convolutional neural network (MRCNN) and an adaptive feature recalibration module. In addition, the temporal dependency of the signal can be captured through the temporal context encoder (TCE) of the multi-head attention mechanism.

- MISC (Niknazar & Mednick, 2024) integrates domain knowledge and data-driven methods through a multi-level structural design. The model injects expert knowledge into deep neural networks to improve the interpretability and performance of the model.

### A.6. The Implementation Details and Evaluation Metrics

**Multi-view Classification.** We briefly introduce the details of the experiment. We utilize fully connected networks with a ReLU layer to extract view-specific evidence. The Adam optimizer is used to train the network, where L2-norm regularization is set to $1e^{-5}$. We employ 5-fold cross-validation to select the learning rate from the options of $3e^{-3}$. In all datasets, 20% of the instances are allocated as the test set. The average performance is reported by running each test case five times.

**Sleep Staging.** To extract the time-frequency input, the window size is set as 12 ms, with 50% overlap between adjacent frames. 256-point FFT is utilized, leading to 129-D in the feature axis. The model is trained on Pytorch platform with a NVIDIA RTX 4090 GPU. We use the Adam optimizer with a batch size of 16 to train the proposed model, and the learning rate is initialized as $3.125e^{-5}$ (0.0005/batch size) and $6.25e^{-4}$ (0.01/batch size) in single-epoch network and multi-epoch network, respectively. In addition, the frequency f of the Gabor kernels was clamped between 0 to 35 Hz. In the training, batch samples were randomly selected from the train subset with normalized probability equal to inverse of the number of samples in each class to overcome an unbalanced distribution of classes.

**Evaluation Metrics.** To evaluate the performance of the proposed sleep stage classification method. We use accuracy ($Acc$), macro F1-score ($MF1$), and Cohen's kappa ($Kappa$). They are defined as follows:

$$Acc = \frac{\sum_{stages} TP}{N}, \tag{13}$$

$$Kappa = \frac{\bar{P} - \bar{P}_e}{1 - \bar{P}_e}, \tag{14}$$

$$MF1 = \frac{\sum_{stages} F1}{M}, \tag{15}$$

where $F1 = \frac{2 \times Pre \times Rec}{Pre + Rec}$, $Rec = \frac{TP}{TP + FN}$, $Pre = \frac{TP}{TP + FP}$. Here, $\bar{P}$ denotes the proportion of observed agreements, and $\bar{P}_e$ represents the expected agreement by chance. $N$ denotes the number of all samples and $M$ denotes the number of sleep stage classes. For a specific sleep stage $S$, $TP$ refers to the number of 30-second epochs correctly classified as stage $S$, $FN$ is the number of epochs that truly belong to stage $S$ but are misclassified as other stages, and $FP$ denotes the number of epochs incorrectly classified as stage $S$ when they actually belong to a different stage.

### A.7. The Quantification of Gabor Kernel Influence

Since the output of the Gabor convolutional layer (GCL) can not be directly used to estimate the contribution of each Gabor kernel to the final decision, we compute the sensitivity of the decision layer output with respect to the output of the Gabor convolutional layer and use it as the normalization weights. Specifically, the sensitivity is quantified by computing the local gradient of the decision output $O[class]$ with respect to the time series output of the $i$th GCL filter, defined as:

$$Sen(t)_{GCL^i \to O[class]} = \frac{dO[class]}{dGCL^i(t)}. \tag{16}$$

This sensitivity reflects how changes in the GCL output at time $t$ influence the decision. We compute the functional effect of each kernel by combining its output with the corresponding positive sensitivity (Ancona et al., 2019). Then the positive contribution of each Gabor kernel is quantified as the squared sum of its output modulated by the corresponding positive sensitivities:

$$Eff^{X_t}_{GCL^i \to O[class]}(t) = GCL^i(t)Sen(t)_{GCL^i \to O[class]}$$
$$\times \theta(Sen(t)_{GCL^i \to O[class]}), \tag{17}$$

$$\overline{Eff}^{X_t}_{GCL^i \to O[class]} = \sum_t (Eff^{X_t}_{GCL^i \to O[class]}(t))^2, \tag{18}$$

where $\theta(\cdot)$ is the Heaviside step function, and $\overline{Eff}^{X_t}_{GCL^i \to O[\text{class}]}$ ($\overline{Eff}$) denotes the positive functional effect of the $i$th Gabor kernel on the $class$th element of the output given input $X_t$ (EEG). By averaging $\overline{Eff}$ across all test samples, the overall impact of each kernel on sleep stage classification is summarized by $Eff_i$.

$$Eff_i = \sum_j \frac{1}{N_j} \sum_{X_t} \overline{Eff}^{X_t}_{GCL^i \to O[class]} \delta(class_X - j), \tag{19}$$

where $N_j$ is total number of the test epochs in sleep stage $j$, $\delta(\cdot)$ is the unit impulse function and $O[class_X]$ is the real sleep stage of the relative input signals $X_t$. $Eff_i$ which is the average of the positive functional effect on the real output class, can represent the overall qualitative impact of each of the Gabor kernels on the decision making process.

### A.8. Supplementary Figures

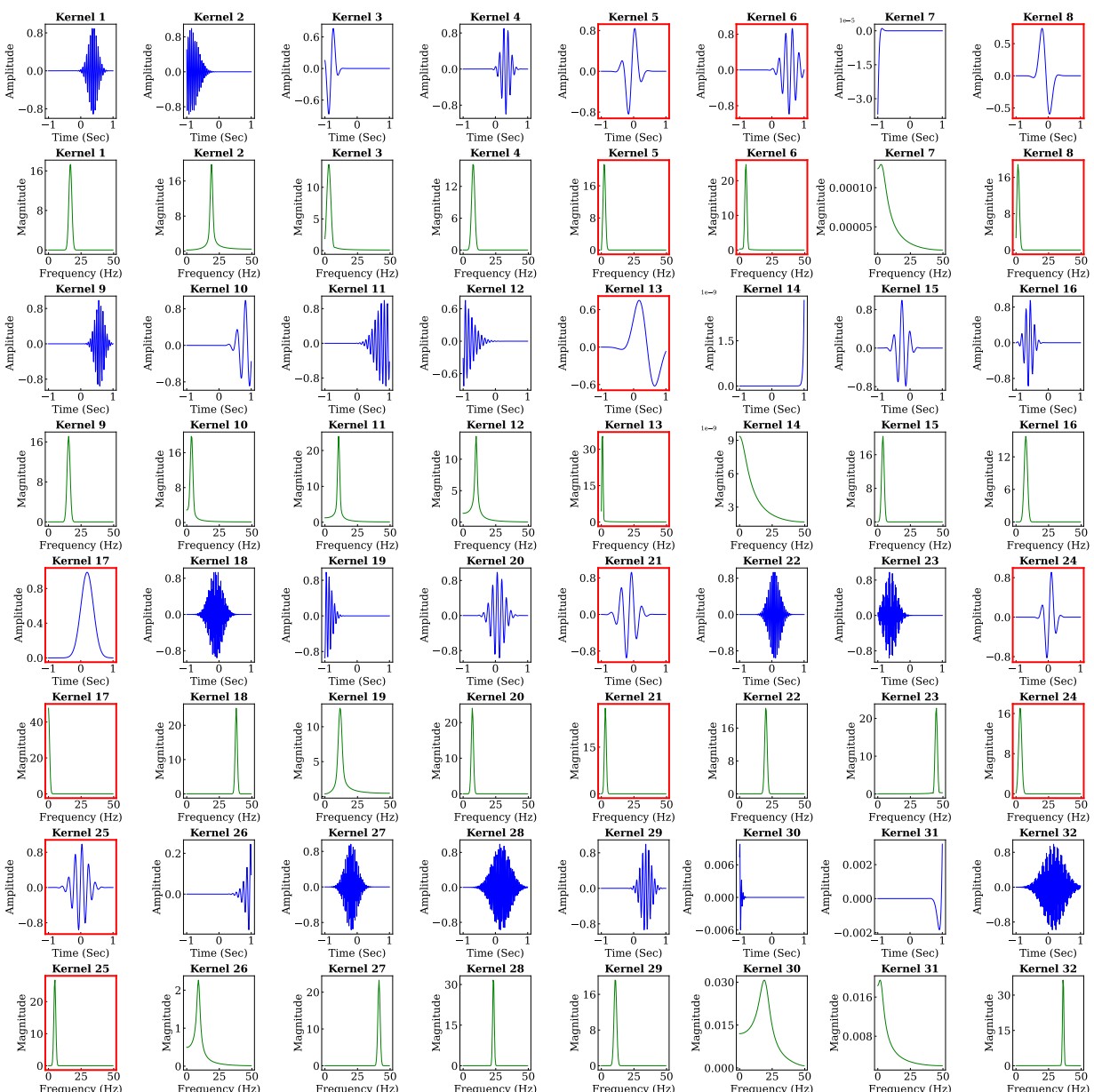

*Figure 8.* Waveform and frequency domains of all optimized Gabor kernels.

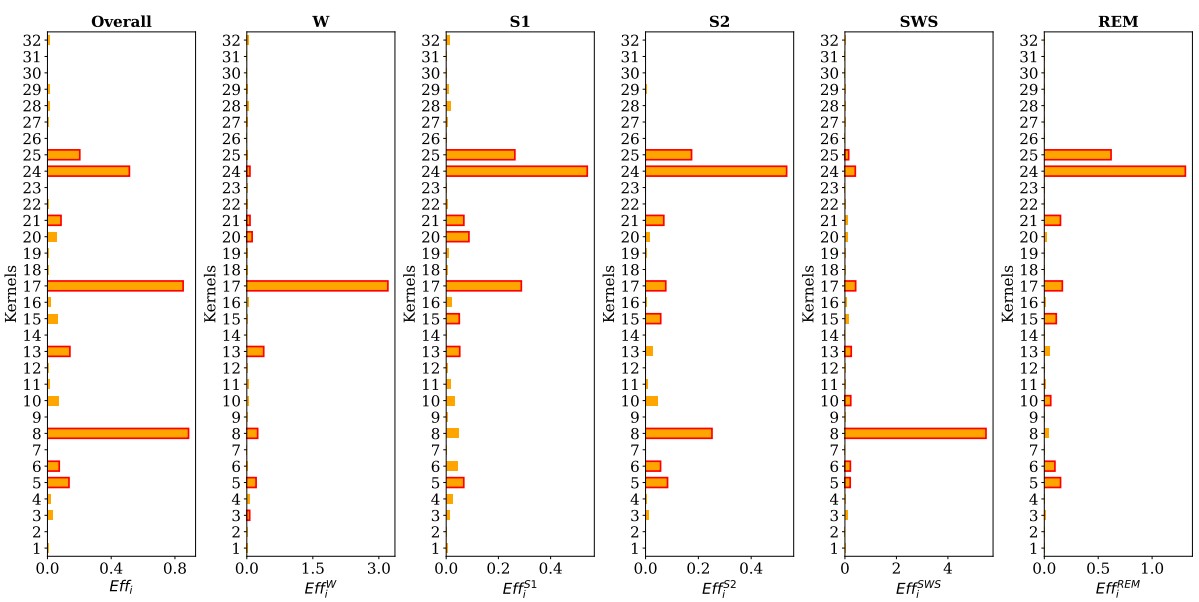

*Figure 9.* Impact of the Gabor kernels on sleep stage scoring.

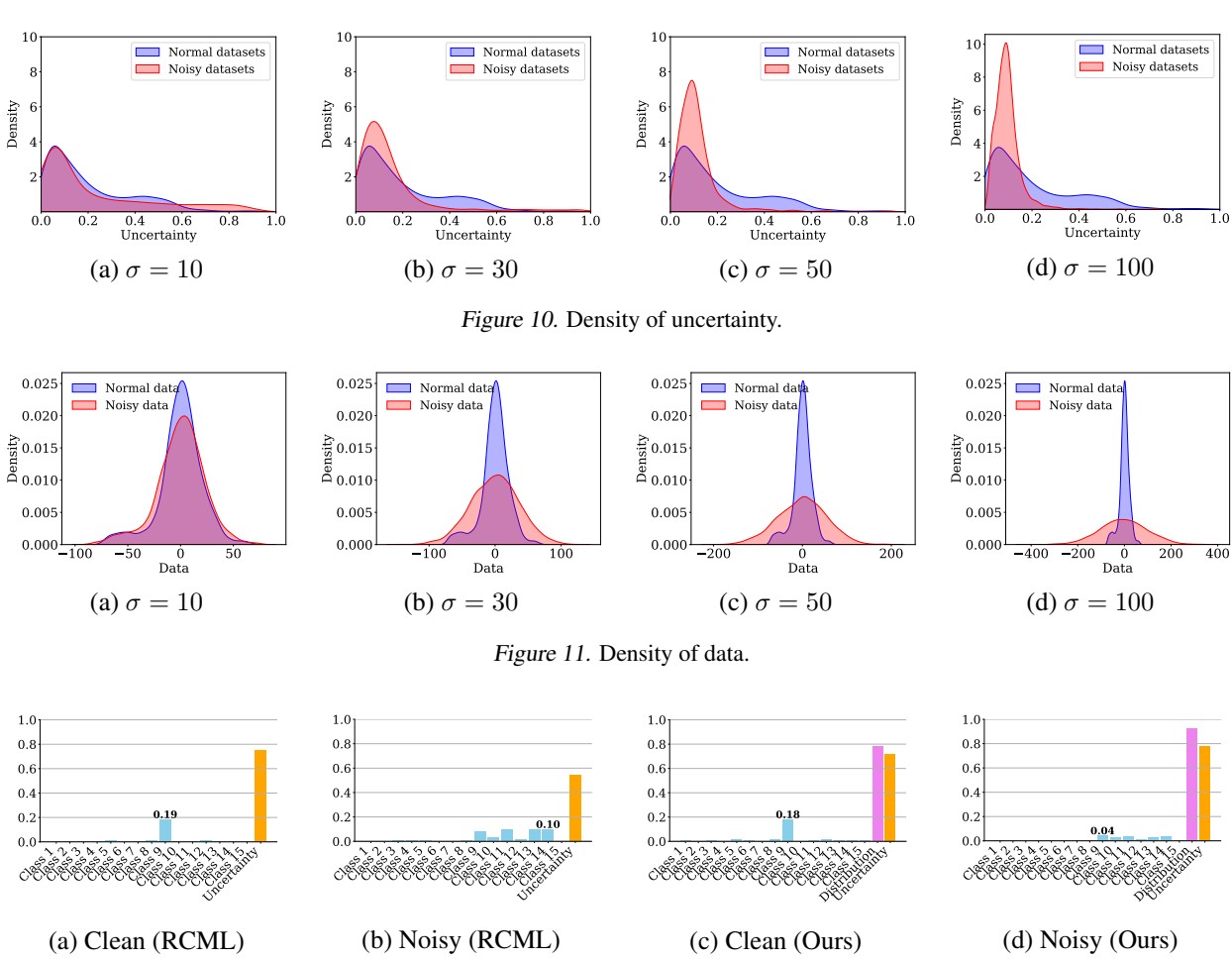

*Figure 10.* Density of uncertainty.

*Figure 11.* Density of data.

*Figure 12.* Comparison of our method and RCML under one sample of Scene15.

