# OpenReview forum: "Trusted Multi-View Classification  with Expert Knowledge Constraints"
_ICML.cc/2025/Conference — ICML 2025 spotlightposter_

### Official Review · Reviewer_x8HD · 2025-03-04

**Overall Recommendation:** 4

**Summary:**

This paper proposes Trusted Multi-View Classification with Expert Knowledge Constraints (TMCEK). There are core contributions: (1) Integrating expert knowledge into multi-view learning to enhance both interpretability and uncertainty estimation, and (2) proposing a novel distribution-aware subjective opinion framework that extends the conventional model through the incorporation of an evidence distribution concentration measure. Finally, the effectiveness of TMCEK is validated on sleep stage classification, outperforming existing models in both classification accuracy and interpretability.

**Claims And Evidence:**

Yes

**Essential References Not Discussed:**

The authors have cited the essential related works for understanding the key contributions of the paper.

**Experimental Designs Or Analyses:**

The experimental setup is comprehensive, with evaluations on multiple datasets and comparisons against a variety of baseline models. The experiments are well-designed, but the testing data distribution and the specifics of how the datasets were divided could benefit from more clarity.

**Methods And Evaluation Criteria:**

Yes. The proposed method and evaluation criteria make sense for the problem.

**Other Comments Or Suggestions:**

Some symbols like d lack a proper explanation in the context.

**Other Strengths And Weaknesses:**

Strengths:
1.The framework is innovative, combining interpretability and performance improvement.
2.The use of expert knowledge through Gabor kernels enhances model transparency.
3.The promising results, particularly in the context of sleep disorder diagnosis, are crucial in areas where trust and transparency are essential. The paper's strong demonstration of the method's performance across multiple datasets further adds to its impact.
4.The paper is well-written and structured, offering a clear and thorough explanation of both the methodology and results.
Weaknesses:
1.The paper presents the method primarily for sleep staging, but there is limited discussion on how this method might generalize to other domains.
2.The paper compares the proposed TMCEK model to various existing multi-view methods like EDL and RCML. Could you provide more details on how the distribution-aware subjective opinion mechanism improves uncertainty estimation compared to traditional methods?

**Questions For Authors:**

1.The paper presents the method primarily for sleep staging, but there is limited discussion on how this method might generalize to other domains.
2.The paper compares the proposed TMCEK model to various existing multi-view methods like EDL and RCML. Could you provide more details on how the distribution-aware subjective opinion mechanism improves uncertainty estimation compared to traditional methods?

**Relation To Broader Scientific Literature:**

The paper contributes to the broader field of multi-view learning and uncertainty estimation. It builds on previous work in trusted learning and expert knowledge integration. The authors effectively position TMCEK within the context of related literature, referencing key studies on multi-view learning, evidence theory, and sleep stage classification.

**Theoretical Claims:**

I have checked the correctness of Proposition 3.4 and Proposition 3.5. The algebraic steps in Appendices A.1–A.2 are clear and free of errors, demonstrating the framework’s mathematical validity. This theoretical foundation directly supports the empirical results, reinforcing the method’s reliability.

---

> ### Author Rebuttal · Authors · 2025-03-28
>
> Thank you for your constructive and encouraging comments. Below are our responses.
>
> **Q: The paper presents the method primarily for sleep staging, but there is limited discussion on how this method might generalize to other domains.**
>
> A: Thanks for your professoional question. While our method is demonstrated for sleep stage classification, the underlying framework of integrating expert knowledge through Gabor-based feature extraction and a distribution-aware subjective opinion mechanism is **general**.
> In any application where domain-specific patterns are critical (e.g., medical imaging, speech recognition, or remote sensing) , one could parametrize them and tailor the initial kernel functions or expert constraints accordingly.
>
> For example, our framework can be effectively applied to image classification by adapting its core components to process image data. In this setting, the first convolutional layer is replaced with 2D Gabor filters, which are designed to capture edges, textures, and other structural patterns in images. These Gabor kernels can be initialized with parameters informed by expert knowledge—such as preferred orientations, scales, and frequencies that are known to highlight important features in images—and then fine-tuned through backpropagation during training. This results in a feature extraction process that is both interpretable and closely aligned with domain-specific patterns.
> Additionally, our framework employs a distribution-aware subjective opinion mechanism to estimate uncertainty. In image classification, this mechanism quantifies not only the overall evidence supporting each class but also the dispersion of that evidence, allowing for a more reliable estimation of confidence in the predictions.
>
> However, we acknowledge that in certain domains lacking sufficient domain-specific expert knowledge or clearly defined feature representations, our framework may not be fully applicable.
>
> **Q: The paper compares the proposed TMCEK model to various existing multi-view methods like EDL and RCML. Could you provide more details on how the distribution-aware subjective opinion mechanism improves uncertainty estimation compared to traditional methods?**
>
> A: Traditional uncertainty estimation methods, such as those based on Evidential Deep Learning (EDL), rely primarily on the overall sum of evidence, making them insensitive to its distribution across classes. To address this, our distribution-aware subjective opinion mechanism incorporates an explicit measure of evidence concentration using the Gini coefficient. This dual consideration of both magnitude and distribution improves sensitivity by assigning higher uncertainty when evidence is more concentrated, even if the total sum remains constant, and enhances conflict resolution by adjusting the fusion rule to weight each view’s contribution based on evidence dispersion.
>
> Morever, theoretical analysis (see Propositions 3.4 and 3.5 in our paper) and experimental results demonstrate that this approach leads to more reliable confidence estimates, particularly in scenarios with ambiguous or noisy data.

---

### Official Review · Reviewer_YgiK · 2025-03-09

**Overall Recommendation:** 4

**Summary:**

This paper introduces an innovative trusted multi-view classification approach designed to tackle the significant shortcomings of existing methods, namely, opacity at the feature level and imprecise confidence assessments at the decision level. Its primary contribution resides in advancing current trusted multi-view classification techniques by bolstering interpretability at the feature level and refining uncertainty estimation. This enhanced framework is subsequently applied to sleep stage classification tasks. The proposed methodology demonstrates superior performance compared to state-of-the-art (SOTA) methods across multiple sleep stage classification datasets.

**Claims And Evidence:**

Yes.

**Essential References Not Discussed:**

The paper has discussed the essential references.

**Experimental Designs Or Analyses:**

I have verified the robustness and validity of the experimental designs and analyses. The designs encompass a thorough evaluation, including performance benchmarks against previous state-of-the-art (SOTA) methods, assessments of individual module effectiveness, an analysis of hyperparameter sensitivity, interpretability, and robustness.

**Methods And Evaluation Criteria:**

Yes.

**Other Comments Or Suggestions:**

I have not any other comments or suggestions here. see Strengths and Weaknesses

**Other Strengths And Weaknesses:**

Strengths:

--The paper provides enhancements in both performance and interpretability.
--The integration of expert knowledge with uncertainty estimation represents a notable contribution. To my knowledge, this is the pioneering work in applying trusted multi-view classification to sleep stage classification, marking a practical application.

Weaknesses:

--In regards to the lower classification performance observed for the N1 stage (Figure 4), have the authors investigated methods to mitigate class imbalance or improve classification accuracy for this specific stage?
--The paper utilizes the Gini coefficient to measure evidence concentration. Have the authors considered alternative methods, and what potential implications might these alternatives have on the overall results?

**Questions For Authors:**

see weaknesses

**Relation To Broader Scientific Literature:**

The paper is related to trusted multi-view learning‌. Its core contribution lies in ‌enhancing existing trusted multi-view classification through improved feature-level interpretability and uncertainty estimation‌, ‌and extending it to sleep stage classification task.

**Theoretical Claims:**

I have checked the correctness of all proofs for Propositions 3.4 and 3.5 in Sec. A.1 and A.2. They are clearly articulated and mathematically rigorous, leaving little room for doubt about the soundness of the theoretical claims.

---

> ### Author Rebuttal · Authors · 2025-03-29
>
> Thank you for the feedback and suggestions. Below are our responses.
>
>
> **Q: In regards to the lower classification performance observed for the N1 stage (Figure 4), have the authors investigated methods to mitigate class imbalance or improve classification accuracy for this specific stage?**
>
> A: Regarding the lower classification performance observed for the N1 stage, our paper takes sampling strategy to alleviate the challenges posed by the unbalanced distribution of classes (in Appendix A.6). During training, batch samples are randomly selected from the training subset using normalized probabilities that are inversely proportional to the number of samples in each class. This ensures that underrepresented classes such as N1 are more likely to be included in each batch, thereby mitigating class imbalance.
> On the Sleep-EDF20 dataset, we use one fold to verify the effectiveness of the sampling strategy.
> | Method    | acc    | f1     | kappa  | wake_f1 | n1_f1  | n2_f1  | n3_f1  | rem_f1 |
> |----------|--------|--------|-------|--------|-------|-------|-------|-------|
> | Sample   | 0.8674 | 0.7953 | 0.8266 | 0.9254 | 0.4035 | 0.8883 | 0.9246 | 0.8347 |
> | No Sample| 0.8725 | 0.7672 | 0.8312 | 0.9288 | 0.2532 | 0.8943 | 0.9313 | 0.8285 |
>
>
> From the results, we can observe that the sampling strategy effectively mitigates the class imbalance issue by significantly boosting N1 classification performance, leading to a better overall balance (as reflected by the macro F1-score), even though there is a slight reduction in overall accuracy and kappa.
>
> **Q: The paper utilizes the Gini coefficient to measure evidence concentration. Have the authors considered alternative methods, and what potential implications might these alternatives have on the overall results?**
>
> A: Regarding the Gini coefficient used for measuring evidence concentration, we recognize that alternative measures—such as entropy or variance—could be explored. Each measure might bring different sensitivity characteristics in uncertainty estimation. For instance, entropy may offer a more nuanced view of dispersion; however, the theoretical properties of the Gini coefficient, as analyzed in our propositions, provide clear advantages in our framework. Future work will examine these alternatives to assess their potential impact on both uncertainty quantification and overall performance. We replaced the method on the multi-view dataset. In the experiment, we fixed the loss weight $\\beta$ to 0.5 and $\\gamma$ to 0.5. The experimental results are as follows.
> | Method | HD| Scene | CUB | PIE |
> |  ----  | ----  |  ----  | ----  |  ----  |
> | Gini      | 98.40 $\pm$ 0.37 | 72.60 $\pm$ 0.99 | 95.33 $\pm$ 1.25 | 96.47 $\pm$ 0.98 |
> | Var       | 98.15 $\pm$ 0.51 | 72.42 $\pm$ 1.24 | 93.00 $\pm$ 2.56 | 96.76 $\pm$ 1.19 |
> | Entropy| 97.75 $\pm$ 0.65 | 67.09 $\pm$ 1.21 | 94.00 $\pm$ 1.62 | 94.41 $\pm$ 2.01 |
>
> These results suggest that the theoretical properties of the Gini coefficient offer a robust measure for capturing evidence dispersion, which in turn contributes to better uncertainty estimation and overall performance.

---

> > ### Comment · Reviewer_YgiK · 2025-04-04
> >
> > Thanks to the responeses in detail from the authors.  I confirmed that the responses have solved all of my concerns. In consideration of the comments from the other reviewers, I will keep my decision.

---

> > > ### Author Response · Authors · 2025-04-07
> > >
> > > We sincerely appreciate your valuable suggestions and guidance, as well as your thoughtful recognition of our work.

---

### Official Review · Reviewer_YxMg · 2025-03-10

**Overall Recommendation:** 4

**Summary:**

This paper proposed a novel trusted multi-view classification method, called TMCEK. Compared with the existing trusted multi-view classification methods, TMCEK embeds the Gabor function into the first convolutional layer as its kernel to enhance feature-level interpretability. Moreover, it introduces a distribution-aware subjective opinion mechanism to derive more reliable and realistic confidence estimates. The TMCEK obtains state-of-the-art results against compared SOTA methods, especially providing an interpretability feature map aligned with humans.

**Claims And Evidence:**

Yes.

**Essential References Not Discussed:**

The paper has covered and discussed the essential references.

**Experimental Designs Or Analyses:**

The experimental designs for evaluating the proposed model involve five aspects: performance comparison, each module effectiveness, hyper-parameter sensibility, interpretation and robustness. I think that the experimental designs are solid and the results are sufficient for supporting their claims.

**Methods And Evaluation Criteria:**

Yes, the proposed method makes sense for the problem at hand.

**Other Comments Or Suggestions:**

Please See the Weaknesses

**Other Strengths And Weaknesses:**

Strengths:
1. The expert knowledge is applied to the trusted multi-view classification, which is a fresh view to trusted multi-view classification.
2. The work finds a new problem that the subjective opinion is distribution-unaware and defines distribution-aware subjective opinion by incorporating the distribution of evidence.
3. Compared with exiting methods, the paper not only achieves state-of-the-art results, but also provides a confident degree and Gabor kernel aligned with human domain experts for decision results.

Weaknesses:
1. The use of Gabor kernels for feature extraction at the first convolutional layer is a key feature of the model. Could you provide further details on how the Gabor kernels are optimized during training and how they compare to other common feature extraction techniques regarding their impact on sleep stage classification?
2. The paper employs attribution maps to analyze the importance of the Gabor kernels. Clarify and extend the explanation of the saliency map method. A more detailed background on this attribution technique would help readers, particularly those not familiar with this approach, to fully appreciate its significance in your model’s interpretability.

**Questions For Authors:**

Please See the Weaknesses

**Relation To Broader Scientific Literature:**

This work improves the trusted multi-view classification from two aspects: (1) feature-level interpretability by embedding expert knowledge and (2) more reliable and realistic confidence estimate by incorporating the distribution of evidence. These contributions are completely innovative.

**Theoretical Claims:**

I have checked the correctness of all proofs for theoretical claims. The theoretical claims regarding the uncertainty estimation mechanism are solid and well-supported by mathematical analysis. The authors present a rigorous derivation that clearly shows how their distribution-aware subjective opinion framework enhances uncertainty quantification compared to traditional methods. The proofs provided for Propositions 3.4 and 3.5 are mathematically sound and logically consistent. Overall, there are no apparent issues with the correctness of these proofs.

---

> ### Author Rebuttal · Authors · 2025-03-28
>
> We are sincerely grateful to the reviewer for dedicating their time and effort to review our work. Below are our responses.
>
> **Q: The use of Gabor kernels for feature extraction at the first convolutional layer is a key feature of the model. Could you provide further details on how the Gabor kernels are optimized during training and how they compare to other common feature extraction techniques regarding their impact on sleep stage classification?**
>
> A: We are sincerely grateful to the reviewer for dedicating their time and effort to review our work. The Gabor kernels are embedded in the first convolutional layer, where their parameters are jointly optimized with the rest of the network via backpropagation. Their design aligns with expert knowledge of EEG waveforms, with some kernels learning to match critical patterns such as slow waves, delta, or theta rhythms that are key in sleep staging.
>
> In contrast to standard convolutional kernels that are learned from scratch without any explicit domain bias, Gabor kernels offer enhanced interpretability—since their waveform shapes can be directly compared to known EEG characteristics—and guided feature extraction by acting as filters tuned to critical frequency bands.
>
> Experimental results indicate that this guided approach not only yields competitive classification performance but also improves the model’s interpretability, as demonstrated by our attribution analysis.
>
> **Q: The paper employs attribution maps to analyze the importance of the Gabor kernels. Clarify and extend the explanation of the saliency map method. A more detailed background on this attribution technique would help readers, particularly those not familiar with this approach, to fully appreciate its significance in your model’s interpretability.**
>
> A: We use saliency maps to interpret the role of individual Gabor kernels in decision making by computing the gradient of the output (or evidence) with respect to the feature maps from the Gabor layer. This approach relies on two key ideas: sensitivity analysis, where regions or kernels that cause large changes in the output when perturbed are deemed more important, and visual attribution, which allows us to directly visualize which kernel responses contribute most to a specific classification.
>
> Attribution maps are generated by backpropagating the gradient information through the network, effectively highlighting the areas that have the highest impact on the final decision. This method not only enhances transparency by linking specific kernel activations to decision outcomes but also bridges the gap between black-box deep learning models and expert interpretability requirements.
>
> For readers less familiar with attribution methods, this technique is well-documented (see, e.g., Ancona et al., 2019) and serves as a powerful tool to validate that kernels corresponding to critical EEG patterns—such as slow waves or theta waves—are indeed influential in the sleep staging decision.

---

> > ### Comment · Reviewer_YxMg · 2025-04-03
> >
> > Thanks for the response. The further detailed clarifications have addressed most of my concerns. After reading the comments from other reviewers, I would like to keep my positive rating.

---

> > > ### Author Response · Authors · 2025-04-07
> > >
> > > We sincerely appreciate your valuable suggestions and guidance, as well as your thoughtful recognition of our work.

---

### Official Review · Reviewer_8nsw · 2025-03-12

**Overall Recommendation:** 4

**Summary:**

This study proposes an expert knowledge-guided trusted multi-view classification framework that achieves dual advancements in interpretability and uncertainty quantification. Specifically, the proposed method introduces expert knowledge as a tool of the feature-level interpretability and defines distribution-aware subjective logic for more sensitive uncertainty estimation. The experimental results show its superior performance on three public datasets.

**Claims And Evidence:**

Yes

**Essential References Not Discussed:**

The paper has included the main related works that are crucial for understanding the context and significance of their contributions.

**Experimental Designs Or Analyses:**

I have checked all the experiments in the experimental section.

**Methods And Evaluation Criteria:**

Yes, the proposed method and evaluation criteria make sense for the trusted multi-view classification.

**Other Comments Or Suggestions:**

See weaknesses

**Other Strengths And Weaknesses:**

Strengths:

1. The novel integration of expert knowledge and uncertainty estimation offers clear advantages.

2. The paper provides strong experimental validation and achieves the SOTA results.

3. The deficiency of existing trusted multi-view methods and the superiority of the proposed method is illustrated by example. Overall, the organization of paper is coherent and easily understood.

Weaknesses:
1. Could you clarify why certain Gabor kernel outputs are considered redundant during the training process? Are there any strategies for improving kernel optimization?
2. It would be beneficial to include ablation experiments where different components of the model are removed or modified, such as removing the Gabor layer or other key modules. This would help quantify the contribution of each part of the model to the overall performance.
3. It might be beneficial to further explore the hyperparameters involved in the loss function. Could you provide additional insight or analysis regarding how different hyperparameter settings might impact the performance of the model?

**Questions For Authors:**

See weaknesses

**Relation To Broader Scientific Literature:**

The paper has a close relation to multi-view learning and uncertainty estimation.  It makes a significant contribution at enhancing interpretability and uncertainty estimation.

**Theoretical Claims:**

Based on my check, the theoretical claims made in this paper are underpinned by a solid mathematical foundation. The authors successfully demonstrate, through rigorous proofs in Propositions 3.4 and 3.5, how their novel distribution-aware subjective opinion framework offers improved uncertainty estimation.

---

> ### Author Rebuttal · Authors · 2025-03-30
>
> We appreciate your detailed feedback and thoughtful questions. Below are our responses.
>
> **Q: Could you clarify why certain Gabor kernel outputs are considered redundant during the training process? Are there any strategies for improving kernel optimization?**
>
> A: Some Gabor kernels become redundant during training because the training process could not optimize them or their learned waveforms overlap significantly with others. In our framework, the first convolutional layer uses Gabor kernels whose parameters (center, width, frequency) are tuned via gradient descent. When multiple kernels converge to capture similar EEG characteristics (for example, several may approximate slow‐wave or theta patterns), only a subset is needed to sufficiently cover the most discriminative features. The training loss naturally down‐weights kernels that do not contribute additional information to the final decision, making some kernels effectively redundant.
>
> To improve kernel optimization, strategies such as incorporating diversity regularization can be employed to explicitly penalize similarity among kernels and encourage each to capture distinct patterns. Additionally, using domain-informed initialization schemes that span a broader range of frequencies and scales, as well as regularization techniques like adding penalty terms or employing dropout to discourage redundancy, can promote a more distributed set of feature extractors.
>
> **Q: It would be beneficial to include ablation experiments where different components of the model are removed or modified, such as removing the Gabor layer or other key modules. This would help quantify the contribution of each part of the model to the overall performance.**
>
> A: We agree that ablation studies are essential for quantifying the contribution of each module.
> In our ablation experiments on one fold of the Sleep-EDF 20 dataset, we evaluated the impact of removing the Gabor layer and the Distribution-Aware Fusion Module. The results are as follows:
> | Gabor | Distribution | acc    | f1    | kappa  |
> |-------|-------------|--------|-------|--------|
> | ✔     | ✔           | 0.8673 | 0.7953| 0.8266 |
> | ✔     | ✘           | 0.8613 | 0.7735| 0.8173 |
> | ✘     | ✔           | 0.8618 | 0.7683| 0.8179 |
>
> These results demonstrate that the complete model—integrating both the Gabor layer and the Distribution-Aware Fusion Module—achieves higher overall performance. This indicates that the two components work: the Gabor layer enhances interpretability by learning EEG features aligned with expert knowledge (such as slow, delta, or theta waves), while the Distribution-Aware Fusion Module optimizes uncertainty estimation by considering the distribution of evidence. Together, they not only improve classification accuracy but also make the model’s decision-making process more transparent.
>
> **Q: It might be beneficial to further explore the hyperparameters involved in the loss function. Could you provide additional insight or analysis regarding how different hyperparameter settings might impact the performance of the model?**
>
> A: Our overall loss function is composed of three components. The first component is the accuracy loss computed from the aggregated evidence. The second component is the accuracy loss computed for each individual view, which is then weighted by β. The third component is the consistency loss, which combines two sub-losses that measure the differences in probability outputs across views and the cosine similarity of the evidence, weighted by ζ and η, and then scaled by γ in the overall loss. We conducted hyperparameter experiments on ζ and η as follows.
> | ζ | η | HD | Scene15 | CUB | PIE |
> |  ----  | ----  |  ----  | ----  |  ----  |  ----  |
> | 0.3 | 0.7 | 97.70$\pm$0.37 | 73.47$\pm$1.44 | 92.50$\pm$2.17 | 96.32$\pm$1.04 |
> | 0.4 | 0.6 | 98.00$\pm$0.47 | 72.73$\pm$1.37 | 94.33$\pm$1.33 | 96.47$\pm$1.50 |
> | 0.5 | 0.5 | 98.40$\pm$0.37 | 72.60$\pm$0.99 | 95.33$\pm$1.25 | 96.47$\pm$0.98 |
> | 0.6 | 0.4 | 98.35$\pm$0.51 | 72.60$\pm$1.24 | 94.67$\pm$1.55 | 97.06$\pm$1.80 |
> | 0.7 | 0.3 | 98.05$\pm$0.80 | 73.22$\pm$1.34 | 93.67$\pm$1.80 | 95.59$\pm$1.32 |
>
> Based on the experimental results, we can observe that the values of ζ and η do not have a significant impact on the overall performance. This indicates that while these hyperparameters play a role in balancing the consistency loss components, their precise values can be flexibly chosen according to specific application requirements without drastically affecting the results.

---

> > ### Comment · Reviewer_8nsw · 2025-04-08
> >
> > Thanks for your response. My concerns have been addressed. I also read your replies to the other reviewers, and I think this is a good work and can contribute to the field of Trusted Multi-View Classification. I'm happy to raise my score to 4.

---

> > > ### Author Response · Authors · 2025-04-08
> > >
> > > We sincerely appreciate your thoughtful recognition of our work and raise the score. Thanks very much!

---

### Decision · Program_Chairs · 2025-05-01

**Decision:**

Accept (spotlight poster)

**Comment:**

This paper proposes a framework that enhances multi-view classification by integrating expert knowledge (via Gabor filters) and a novel distribution‐aware subjective opinion mechanism. The proposed method is motivated by theoretical proofs and is validated on sleep stage classification datasets. All four reviewers provided an overall recommendation of 4 (Accept), with their feedback converging on the novelty of the contributions, the clarity of the experimental design, and the potential impact on both interpretability and uncertainty estimation in safety-critical applications.

Overall, this work represents an advancement in both trusted multi-view classification and interpretable uncertainty estimation, with potential impact on safety-critical applications. Based on the strength of the contributions, the comprehensive experimental validation, and the satisfactory addressing of all major concerns during the rebuttal phase, the meta-review recommends acceptance of the submission. Minor revisions are encouraged to further refine the kernel optimization explanations and broaden the generalizability discussion.